# Analysis of Regional Satellite Clock Bias Characteristics Based on BeiDou System

Wenxuan Liu [1], Hu Wang [1,*], Hongyang Ma [2], Yingyan Cheng [1], Pengyuan Li [3], Bo Li [4] and Yingying Ren [3]

1. Chinese Academy of Surveying & Mapping, Beijing 100036, China
2. School of Geomatics Science and Technology, Nanjing Tech University, Nanjing 210037, China
3. College of Surveying and Geo-Informatics, Tongji University, Shanghai 200092, China
4. School of Geomatics, Liaoning Technical University, Fuxin 123000, China
* Correspondence: wanghu@casm.ac.cn; Tel.: +86-15652077998

**Abstract:** With the continuous development of the Global Navigation Satellite System (GNSS), the calculation theory and strategy of the global Satellite Clock Bias (SCB) tends to be mature. However, in some eventualities with restricted conditions, the calculation and application of the global SCB are limited; hence, the application of regional SCB is derived. This paper focuses on the quality of regional SCB products in different regions, calculates three groups of regional SCB products, and analyzes their properties and application effects. We expand the double-differenced assessment method for SCB and extend satellite clock accuracy assessment to regional satellite clock products. Additionally, the Regional Effect Bias (REB) is introduced to analyze the influence of the relative position of satellite geometry on the SCB products due to the regional effects. The conclusions are as follows: (1) In low-latitude regions, SCB products have a high degree of completeness and a large number of satellite observations, which is conducive to expanding the positioning application range of regional SCB; (2) the low-latitude regions SCB will be affected by ionospheric activity, and the accuracy will be slightly lower than that of satellite clocks deviation in mid-latitudes; (3) in this paper, the REB in this area is in the level of $10^{-7}$. The experiment displays the result that the values of REB in low-latitude areas are larger, leading to fluctuated Precise Point Position (PPP) results. However, there are fewer stations in the mid-latitude regions, which will also affect the accuracy of PPP; (4) the accuracy of the positioning results of the regional satellite clock deviation in the Chinese region is higher than that of the global clock.

**Keywords:** precise point positioning; clock estimation; station network



## 1. Introduction

After the great success of the Global Positing System (GPS) and Russian Global'naya Navigatsionnaya Sputnikova Sistema (GLONASS), Europe and China have established the Galileo Satellite Navigation System (Galileo) and BeiDou Navigation Satellite System (BDS) separately. In addition to the above four Global Navigation Satellite Systems (GNSS), India and Japan have also established their regional satellite systems, namely, the Indian Regional Navigation Satellite System (IRNSS) and Quasi-Zenith Satellite System (QZSS). The construction of China's BeiDou satellite navigation system is based on a three-step strategy. The BeiDou-1 system (BDS-1) was completed at the end of 2000 to provide services to the east Asia region; the BeiDou-2 system (BDS-2) was completed at the end of 2012 to provide services to the Asia-Pacific region, and the BeiDou-3 system (BDS-3) was completed in 2020 to provide services to the globe.

The compensation of Satellite Clock Biases (SCB) is an important part of GNSS high-precision data processing. Currently, several Analysis Centers (ACs) around the world provide a variety of GNSS clock bias products, e.g., post-processing SCB products, which play a key role in Positioning, Navigation, and Timing (PNT) [1]. The application of post-processing PPP in scientific research includes geodesy, atmospheric monitoring [2], and

plate drift monitoring [3]. In recent years, with the improvement of ultra-rapid satellite orbit accuracy, orbit determination accuracy has met the need for real-time positioning [4]. With the expanded real-time application scenario, the real-time precise point positioning (RT-PPP) technology based on the real-time clock [5–8] has been applied in vehicle navigation [9], autonomous driving, aerial triangulation [10], time transfer [11], deformation monitoring [12], and other fields. Therefore, real-time SCB has become an important research area.

The International GNSS Service (IGS) usually uses fixed orbits and known station coordinates to calculate the post-processing SCB, and the posterior SCB accuracy can reach 0.1 ns [13]. Some ACs use Orbit Determination and Time Synchronization (ODTS) method, for which the orbit product and the SCB product are calculated simultaneously [14]. However, the resulting post-processing SCB product contains the assimilated orbit error [15]. Common methods of studying real-time SCB include the undifferenced method (UD), epoch-differenced method (ED), undifferenced range, and epoch-differenced phase mixed-difference model (MD). The mixed-difference model can eliminate a large number of ambiguity parameters, and the dimension of the obtained equation matrix is small, which can meet the calculation efficiency requirement of real-time estimation [16]. However, the MD method requires additional clock deviation reference. The UD method is to estimate the satellite clock bias, receiver clock bias, atmospheric delay error, and ambiguity parameters simultaneously, and the accuracy of the SCB obtained in this way has higher accuracy, but the ambiguity parameters need to be computed for each epoch and the data calculating efficiency is inefficient [17]. Li implemented a real-time precision clock correction estimation algorithm based on undifferenced carrier phase observations and introduced clock error reference while improving the calculation speed of SCB [18].

To provide RT-PPP services, it is necessary to estimate the precise satellite clocks and orbits quickly and publish them to users [19–21]. Yan et al. applied a regional Continuous Operational Reference System (CORS) service method for SCB estimation [22]. The method is based on ground-tracking station networks covering a region. The system control center receives the real-time observation data from each station and calculates the SCB products precisely. In regional CORS networks using low-cost receivers, PPP using uncombined GNSS observations with ionospheric delay parameters obtained from CORS has good results [8]. However, with the system expansion of the CORS network, the parameters, including phase ambiguity, receiver clock bias, and station Zenith Tropospheric Delay (ZTD), additionally increase. This will reduce computational efficiency. Pan et al. used the broadcast ephemeris to estimate the SCB together with the orbit through the regional CORS station, to realize the regional PPP completely autonomously [23].

In summary, the current research on regional clocks focuses on calculating regional SCB based on regionally distributed CORS to expand the calculation and application scenarios of PPP under regional conditions. However, there are few studies on the characteristics of regional SCB in different regions. In this paper, we fix the orbit products and calculate different regional SCB products by different regional stations in China. The real-time SCB between the global and regional networks in China is computed by the mixed-differenced method (MD). Firstly, this paper counts the epoch which can calculate the SCB of each satellite in different regions within a day to determine whether the data of the SCB of each region can approach the application requirements of PPP. Secondly, the calculated regional SCB and global SCB products are used to evaluate the accuracy of each regional satellite clock. Then, the quadratic fitting coefficient of global SCB is used to study the influence of station region distribution on satellite clock products. Finally, this paper selects stations in different regions and uses PPP to verify the SCB products' quality in each region.

This paper studies the calculation of real-time BeiDou satellite clock deviation in regional networks and analyzes the characteristics of SCB products in the regional network. It expands the application of the BeiDou system and compensates for the previous deficiency of only using the global station net to estimate the SCB products. We study the regional SCB products and their PPP performance. This article is organized as follows. The Section 1

is the introduction, which briefly describes the research status of SCB and the research content of this paper. In Section 2, we briefly introduce the mixed-difference method of SCB calculation and the evaluation strategy of the regional SCB. In Section 3, we calculate the completeness of three different regional SCB products and evaluate their accuracy. In Section 4, we conduct PPP experiments with regional SCB and draw experimental results, and the relevant discussions and conclusions are presented in Sections 5 and 6 separately.

## 2. Data Process

### 2.1. SCB Calculation

The regional SCB in our paper is estimated by mixing undifferenced range and epoch-differential phase, which is usually used for global rapid SCB calculation of GPS satellites [8]. The epoch-difference method is used to remove the ambiguity parameters, and the clock deviation can be corrected by the initial clock deviation estimated by the range. The combination of the two steps can improve the efficiency of the solution on the premise of maintaining accuracy.

The definition equation of the MD model is:

$$\delta t(i) = \delta t(i_0) + \sum_{j=i_0+1}^{i} \Delta \delta_t(j) \tag{1}$$

The clock changes and therefore the accumulated clock corrections $\delta t(i)$ at the epoch number $i$ can be estimated rather precisely; they are biased by the initial clock offset $\delta t(i_0)$ at the starting epoch $i_0$; $\Delta \delta t(j)$ is the cumulative value of the clock bias; the $\Delta$ is the difference operator two adjacent epochs; and the epoch-difference clock $\Delta \delta t(i)$ can be expressed as:

$$\Delta \delta t(i) = \delta t(i) - \delta t(i-1) \tag{2}$$

In order to make the formula concise, we remove the true value of the satellite-receiver distance as the observation value from both sides of the equations of Equations (3) and (4). The observation equations of undifferenced range and epoch-difference phase are:

$$v_{Pc}(i) = \delta_{t_r}(i) - \delta_{t_s}(i) + m(i)\delta T(i) + \varepsilon_{Pc}(i) \tag{3}$$

$$v_{\Delta Lc}(i) = \Delta \delta_{t_r}(i) - \Delta \delta_{t_s}(i) + \Delta m(i)\delta T(i) + \Delta \varepsilon_{\Delta Lc}(i) \tag{4}$$

Since the phase observation value in Equation (4) adopts the epoch-difference phase, the phase observation value and other parameters related to the epoch-difference are denoted by $\Delta$. Where $v_{Pc}$, $\varepsilon_{Pc}$ are the range observation and the residual of the range observation, respectively; $v_{\Delta Lc}$, $\Delta \varepsilon_{Lc}$ are, respectively, the observation of phase and the phase residual of the observation; $\delta T$ and $m$ represent Zenith Total Delay (ZTD) and its mapping function, respectively; $\delta_{t_r}$ and $\delta_{t_s}$ are receiver and satellite clock biases.

Substituting the receiver clock bias $\delta_{t_r}(i)$ and satellite clock bias $\delta_{t_s}(i)$ in Equation (3) by Equation (1), accumulated from epoch $i_{r0}$ and $i_{s0}$, we have:

$$\begin{aligned} v_{Pc}(i) = \Delta \delta_{t_r}(i) - \Delta \delta_{t_s}(i) + m(i)\delta T(i) + \delta_{t_r}(i_{r0}) - \delta_{t_s}(i_{s0}) + \varepsilon_{Pc}(i) \\ + \sum_{j=i_{r0}+1}^{i-1} \Delta \delta t_r(j) - \sum_{j=i_{s0}+1}^{i-1} \Delta \delta t_s(j) \end{aligned} \tag{5}$$

The last two terms of the above equation can be replaced by the clock deviation estimated at the previous epochs; the sum of the last three terms of the distance observation equation is expressed as $\bar{\varepsilon}_{p_c}(i)$. The Equation (5) can be expressed as:

$$v_{Pc}(i) = \Delta \delta_{t_r}(i) - \Delta \delta_{t_s}(i) + m(i)\delta T(i) + \delta_{t_r}(i_{r0}) - \delta_{t_s}(i_{s0}) + \bar{\varepsilon}_{p_c}(i) \tag{6}$$

In this paper, Equations (4) and (6) are used to estimate the initial clock deviation of each station and satellite.

The epoch-difference method can be used to accurately estimate the clock bias variation and the ZTD [15,24,25]. Therefore, when the mixed algorithm computes the clock bias, Equation (4) is first used to estimate the clock biases and ZTDs of each epoch, and then these estimates will be used to correct the range observations so that only the initial clock deviations remain in the range observations. The corresponding observation equations can be obtained from Equation (5) by putting the clock offset parameters at epoch $i$ into the accumulated clocks as follows:

$$v_{Pc}(i) = \delta_{t_r}(i_{r0}) - \delta_{t_s}(i_{s0}) + m(i)\delta T(i) + \varepsilon_{Pc}(i) + \sum_{j=i_{r0}+1}^{i} \Delta \delta t_r(j) - \sum_{j=i_{s0}+1}^{i} \Delta \delta t_s(j) \quad (7)$$

When calculating the initial value, there are no accumulated terms in Equation (7), so we remove the last two terms. The third and fourth terms of Equation (7) are represented by $\widetilde{\varepsilon}_{Pc}(i)$; then, Equation (7) can be expressed as:

$$v_{Pc}(i) = \delta t_r(i_{r0}) - \delta t_s(i_{s0}) + \widetilde{\varepsilon}_{Pc}(i) \quad (8)$$

Through the observation equations established by Equations (4) and (8), the least squares method is used to estimate the SCB of the observed data. The differenced clock and ZTD parameters are estimated using the epoch-differenced phases with Equation (4), and the initial clock biases are estimated using the undifferenced ranges with Equation (8).

### 2.2. SCB Evaluation

Due to the curvature of the earth and the restrictions of regional location, it is difficult for the regional network stations to completely receive the observation signals of the global satellites as the global network. When the satellite's trajectory vanishes from the network's sky, that can result in the interruption and also the absence of the estimation of the satellite clock error product during this period. Therefore, the estimation of most satellite clock products is incomplete, and only some periods' SCB products exist. Therefore, this paper first evaluates whether the completeness of the BeiDou satellite clock product will meet the requirements of PPP.

After the evaluation of the completeness, we compare the product accuracy of the BeiDou SCB. When evaluating the accuracy of the clock product, it is necessary to eliminate the time-scale differences; that is, the deviation of the system from the reference clock product [26,27]. The time scale difference comes from the base clock selected by the system. The SCB product model can be expressed as [15]:

$$C_a^S = O_a + O_a^S + T^S + R_a^S + \varepsilon_a^S \quad (9)$$

In Equation (9), the superscript S and the subscript a represent the satellite and the analysis center, respectively; $C_a^S$ is the clock product; $O_a$ is the time scale differences introduced by the reference clock when calculating the clock bias. $O_a^S$ is the initial clock bias; it is involved in the across-time clock correction between epochs. To restore the satellite clock correction, we must introduce an initial clock. With the limited accuracy of the initial clock, there is a systematic bias between the introduced and theoretical initial clock. The systematic bias is called the initial clock bias $O_a^S$. $T^S$ is the phase estimation clock correction [15]; $R_a^S$ is the effect caused by the assimilated orbital error [28]. Due to the correlation of the orbit and clock offset, most of radial orbital errors can be absorbed by the clock offset. These radial orbit errors are denoted as $R_a^S$ in the clock product; $\varepsilon_a^S$ represents the noise.

In this paper, we use the Single Satellite method (SSM); that is, a satellite is selected to construct inter-satellite differences (SD) to eliminate the time scale differences [29]. $S_0$ is selected as the reference star for inter-satellite difference:

$$
\begin{aligned}
C_{a_1}^{\Delta S} = C_{a_1}^{S} - C_{a_1}^{S_0} \quad &= \left(O_{a_1} + O_{a_1}^{S} + T^{S} + R_{a_1}^{S} + \varepsilon_{a_1}^{S}\right) - \left(O_{a_1} + O_{a_1}^{S_0} + T^{S_0} + R_{a_1}^{S_0} + \varepsilon_{a_1}^{S_0}\right) \\
&= \left(O_{a_1}^{S} - O_{a_1}^{S_0}\right) + \left(T^{S} - T^{S_0}\right) + \left(R_{a_1}^{S} - R_{a_1}^{S_0}\right) + \left(\varepsilon_{a_1}^{S} - \varepsilon_{a_1}^{S_0}\right) = O_{a_1}^{\Delta S} + T^{\Delta S} + R_{a_1}^{\Delta S} + \varepsilon_{a_1}^{\Delta S}
\end{aligned}
\tag{10}
$$

In Equation (10), $\Delta$ is the difference operator, $S$ is the satellite to be evaluated, $S_0$ is the reference satellite, $\Delta S$ represents the difference between satellites and reference satellite, and $O_a$ is the time scale differences. After time scale differences are eliminated, the product-differenced (PD) between different analysis centers is:

$$
\begin{aligned}
C_{\Delta a}^{\Delta S} = C_{a_2}^{\Delta S} - C_{a_1}^{\Delta S} \quad &= \left(O_{a_2}^{\Delta S} + T^{\Delta S} + R_{a_2}^{\Delta S} + \varepsilon_{a_2}^{\Delta S}\right) - \left(O_{a_1}^{\Delta S} + T^{\Delta S} + R_{a_1}^{\Delta S} + \varepsilon_{a_1}^{\Delta S}\right) \\
&= \left(O_{a_2}^{\Delta S} - O_{a_1}^{\Delta S}\right) + \left(R_{a_2}^{\Delta S} - R_{a_1}^{\Delta S}\right) + \left(\varepsilon_{a_2}^{\Delta S} - \varepsilon_{a_1}^{\Delta S}\right) = O_{\Delta a}^{\Delta S} + R_{\Delta a}^{\Delta S} + \varepsilon_{\Delta a}^{\Delta S}
\end{aligned}
\tag{11}
$$

In Equation (10), $\Delta a$ represents the difference between products of different analysis centers, $(\cdot)_{\Delta a}^{\Delta S}$ is the second difference operator. The initial clock deviation $O_a^S$ is a constant offset in the continuous arc, which only affects the convergence time of PPP [22]. The assimilated orbit error $R_a^S$ is periodic and the effect can be eliminated by combining with the corresponding orbit products in PPP processing. It can be seen from the above formulas that the SCB product can be decomposed into the constant item, initial deviation, assimilated orbital period item, and noise in the form after double difference.

All SCB products in this paper are calculated by GFZ orbit; therefore, the influence of assimilation orbit error acting on SCB is the same, and the second difference $R_{\Delta a}^{\Delta S}$ is 0. The initial clock offset is a constant, and so is its second difference $O_{\Delta a}^{\Delta S}$. Therefore, the Standard Deviation (STD) can be used to measure the noise level and evaluate the accuracy of BeiDou satellite clock in this paper.

After accurately estimating BeiDou clock products, the phase, frequency, clock drift, and noise parameters of the BeiDou satellite clock can be obtained by quadratic term fitting of the global SCB, as shown in Equation (12):

$$
\Delta C_i = a_0 + a_1(t_i - t_0) + a_2(t_i - t_0)^2 + \varepsilon_i \, , i = 1, 2, 3 \ldots , n
\tag{12}
$$

where $\Delta C_i$ is the SCB data in the epoch $i$; $a_0$, $a_1$, and $a_2$ are the clock offset, frequency offset, and frequency drift separately; $\varepsilon_i$ is the noise fitting residual; $n$ is the number of epochs of the clock difference of the fitting period. Combined with Equation (12) and the least squares method to process the global clock deviation sequence of the fitting period, the clock error, frequency offset, and frequency drift parameters of the satellite clock in this period can be obtained. Substitute these parameters of the corresponding fitting period to $\Delta TC_i = a_0 + a_1(t_i - t_0) + a_2(t_i - t_0)^2$ to obtain the theoretical clock deviation $\Delta TC_i$ at each epoch; by comparing the SCB products in different regions with the theoretical value of the SCB, the influence of the regional station network on the SCB calculation can be studied.

## 3. Product Analysis

The original observation data of 120 MGEX stations that can receive BeiDou observation data from all around the world are selected in this paper to calculate the global SCB. The specific SCB estimation strategy is shown in Table 1.

In this paper, we calculate global SCB in the strategy shown in Table 1. To verify the clock product, we compare the consistency between our global product and the global products of CODE, WHU, and GFZ. The consistency between products is shown in Figure 1. It can be seen from this figure that the consistency level of the SCB products calculated in this paper differs from other analysis centers at about 0.1 ns, which can verify the correctness of the programs and algorithms in this paper.

**Table 1.** Processing strategy and parameter model of SCB.

| Items | Parameters | | Models for SCB Determination |
|---|---|---|---|
| Observation information | Observation | | UD ionosphere-free range and ED ionosphere-free carrier-phase observation |
| | Prior information | | P1: 1.0 m; L1: 0.01 cycle |
| | Elevation mask | | 7° |
| | Observation weight | | p = 1, elev > 30°, p = 2 sin(elev), elev ≤ 30° |
| Correction | Phase rotation correction | | Model Correct [30] |
| | PCO | Satellite | GPS PCO: IGS08.atx [31]; BeiDou GEO PCO:IGS08.atx; BeiDou IGSO/MEO PCO:ESA Mode [32] |
| | | Receiver | GPS, BeiDou PCO: IGS08.atx |
| | PCV | Satellite | GPS PCV: IGS08.atx; BeiDou PCV: corrected |
| | | Receiver | GPS, BeiDou PCV:IGS08.atx |
| | Tides | | Ocean tides; solid earth tides; solid earth pole tides: IERS conventions 2010 |
| Parameters estimation | Relativistic effects | | IERS conventions 2003 |
| | Reference clock | | One satellite clock |
| | Satellite orbit | | Fixed |
| | Station coordinate | | Fixed |
| | Tropospheric delay | | Saastamoinen model + GMF mapping function random-walk process for each epoch |
| | Satellite/receiver clock | | Estimated as white noise |
| | Ambiguity | | Estimated in un-differenced, eliminated by phase epoch-differenced/estimated if exit cycle slips |

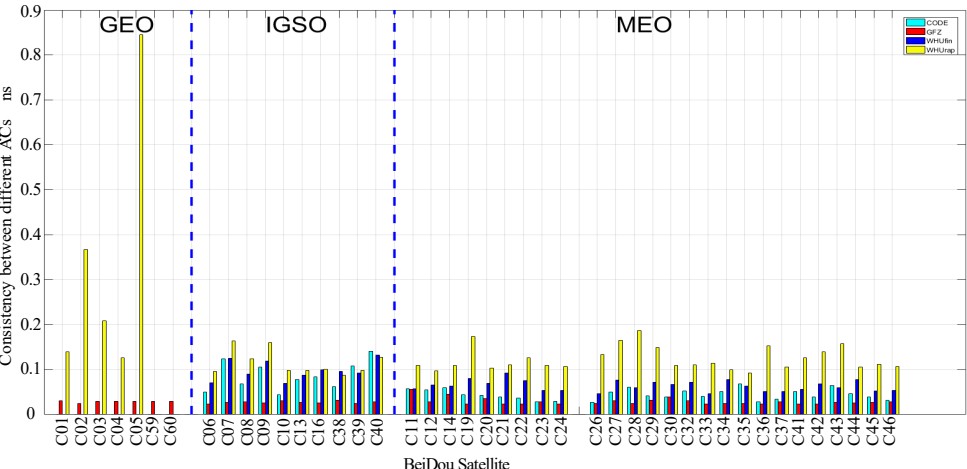

**Figure 1.** The consistency verification.

Three research regions are selected for calculation and SCB, namely, the China regional network, the North China regional network, and the South China regional network. Among 210 national stations distributed nationwide, 50 stations are selected by region to calculate the regional SCB of the BeiDou satellite in each region. The distribution of national stations in the three selected regions is shown in Figures 2–4.

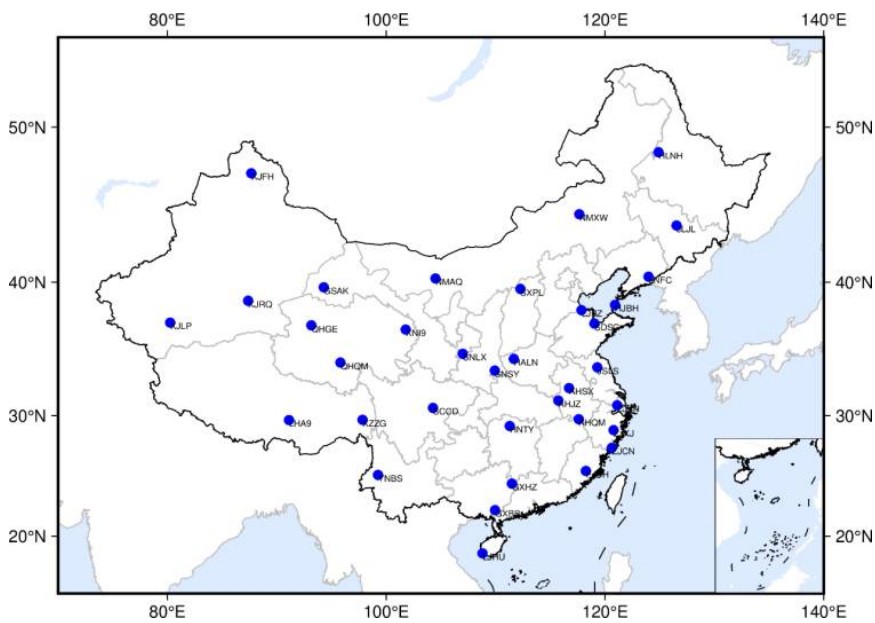

**Figure 2.** Station distribution in China.

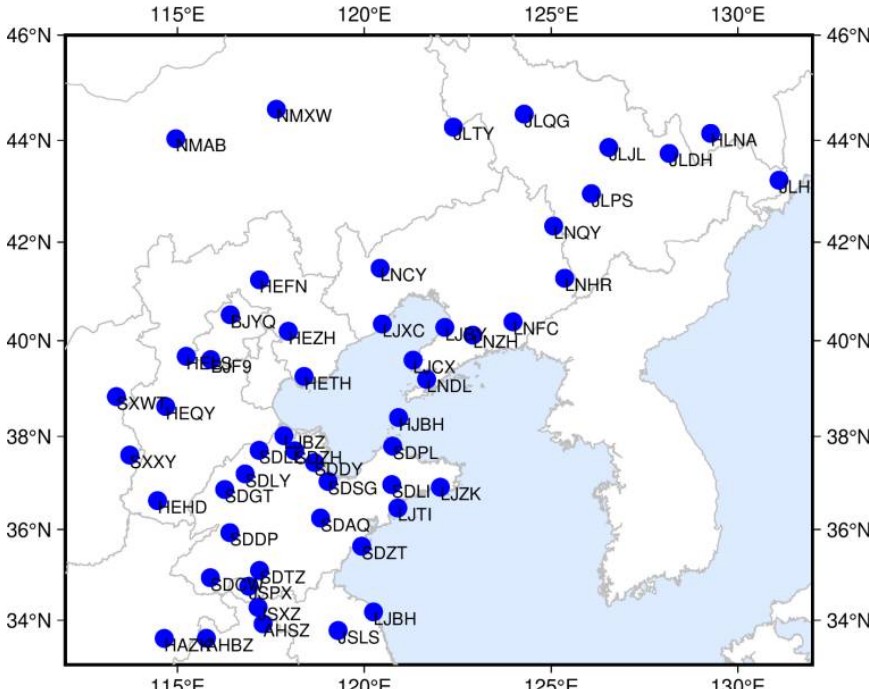

**Figure 3.** Station distribution in North China.

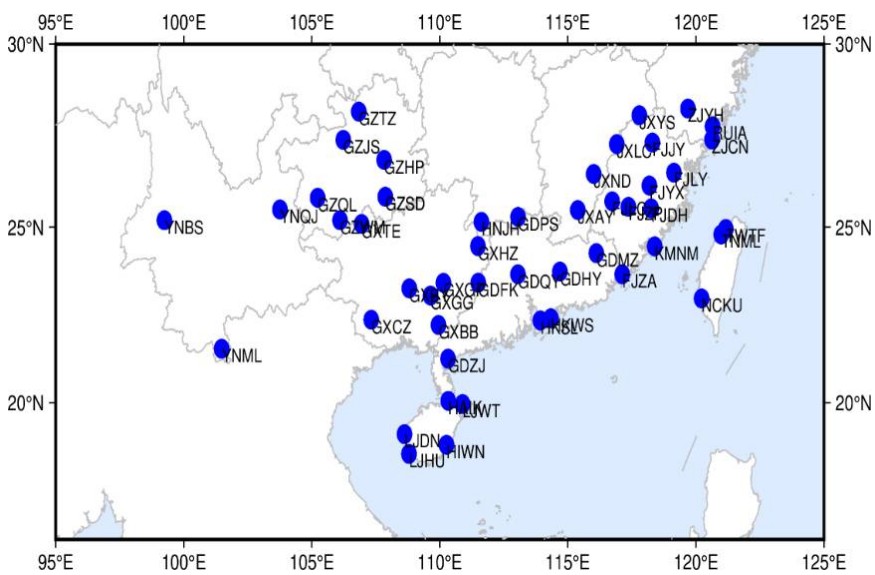

**Figure 4.** Station distribution in South China.

### 3.1. Completeness of Regional Clock Products

Global satellites can be tracked by globally networked stations as they are rotating around the earth, but our application scenario is limited to regionally networked stations. When a satellite is far from the study area, it cannot be tracked by any ground station in that area due to the curvature of the Earth and elevation angle. This phenomenon will result in the satellite clock bias sequence not being able to be obtained completely when only using the observation value of the regional station to calculate the satellite clock bias, causing the satellite clock bias product incompleteness. The most intuitive impact of this incompleteness on the user is the reduction of the number of observations. Too few observations may let users cannot make PPP processing.

Therefore, it is necessary to analyze the completeness of the satellite clock product to study whether it is close to the requirements of PPP. The Geosynchronous Earth Orbit (GEO) and Inclined Geosynchronous Orbit (IGSO) of the BeiDou system's period is 23h56min and the Middle Earth Orbit (MEO)'s period is 12h56min; hence, the satellite-receiver geometry is almost the same every day. The discontinuous situation of regional BeiDou SCB is statistically analyzed by using the SCB product on the day of year 214 in 2021. The results are shown in Tables 2–4. The coverage period diagram is drawn and the results are shown in Figures 5–7 (blue for GEO, yellow for MEO, and red for IGSO).

**Table 2.** Daily coverage of regional SCB data in China.

| | GEO | | IGSO | | | | MEO | | |
|---|---|---|---|---|---|---|---|---|---|
| C01 | 0.00% | C06, C39 | 0.00% | C11 | 51.53% | C25 | 38.19% | C35 | 32.01% |
| C02 | 100.00% | C07 | 95.80% | C12 | 50.21% | C26 | 51.08% | C36 | 26.49% |
| C03 | 100.00% | C08 | 81.94% | C14 | 41.94% | C27 | 27.81% | C37 | 25.38% |
| C04 | 100.00% | C09 | 87.67% | C19 | 39.93% | C28 | 39.83% | C41 | 28.09% |
| C05 | 100.00% | C10 | 94.03% | C20 | 44.06% | C29 | 29.31% | C42 | 33.02% |
| C59 | 100.00% | C13 | 84.55% | C21 | 38.26% | C30 | 27.60% | C43 | 39.90% |
| C60 | 100.00% | C16 | 89.65% | C22 | 36.42% | C32 | 32.05% | C44 | 36.56% |
| | | C38 | 77.74% | C23 | 38.47% | C33 | 29.48% | C45 | 36.46% |
| | | C40 | 76.49% | C24 | 46.63% | C34 | 40.17% | C46 | 17.43% |

**Table 3.** Daily coverage of regional SCB data in North China.

| GEO | | IGSO | | | | MEO | |
|---|---|---|---|---|---|---|---|
| C01 | 0.00% | C06 | 0.00% | C11 | 41.01% | C24 | 31.60% |
| C02 | 100.00% | C07 | 78.30% | C12 | 40.17% | C25 | 33.78% |
| C03 | 100.00% | C08 | 73.26% | C14 | 34.24% | C26 | 39.48% |
| C04 | 100.00% | C09 | 73.89% | C19 | 28.44% | C27 | 21.01% |
| C05 | 41.74% | C10 | 76.15% | C20 | 32.29% | C28 | 27.92% |
| C59 | 0.00% | C13 | 73.09% | C21 | 33.96% | C29 | 27.71% |
| C60 | 0.00% | C16 | 76.08% | C22 | 32.53% | C30 | 22.67% |
| | | C38–C40 | 0.00% | C23 | 32.88% | C32–C37 | 0.00% |
| | | | | | | C41–C46 | 0.00% |

**Table 4.** Comparison of daily SCB coverage between china and the North China region.

| MEO | | | | IGSO | |
|---|---|---|---|---|---|
| C11 | 10.52% | C24 | 15.03% | C07 | 17.50% |
| C12 | 10.03% | C25 | 4.41% | C08 | 8.68% |
| C14 | 7.71% | C26 | 11.60% | C09 | 13.78% |
| C19 | 11.49% | C27 | 6.81% | C10 | 17.88% |
| C20 | 11.77% | C28 | 11.91% | C13 | 11.46% |
| C21 | 4.31% | C29 | 1.60% | C16 | 13.58% |
| C22 | 3.89% | C30 | 4.93% | | |
| C23 | 5.59% | | | | |

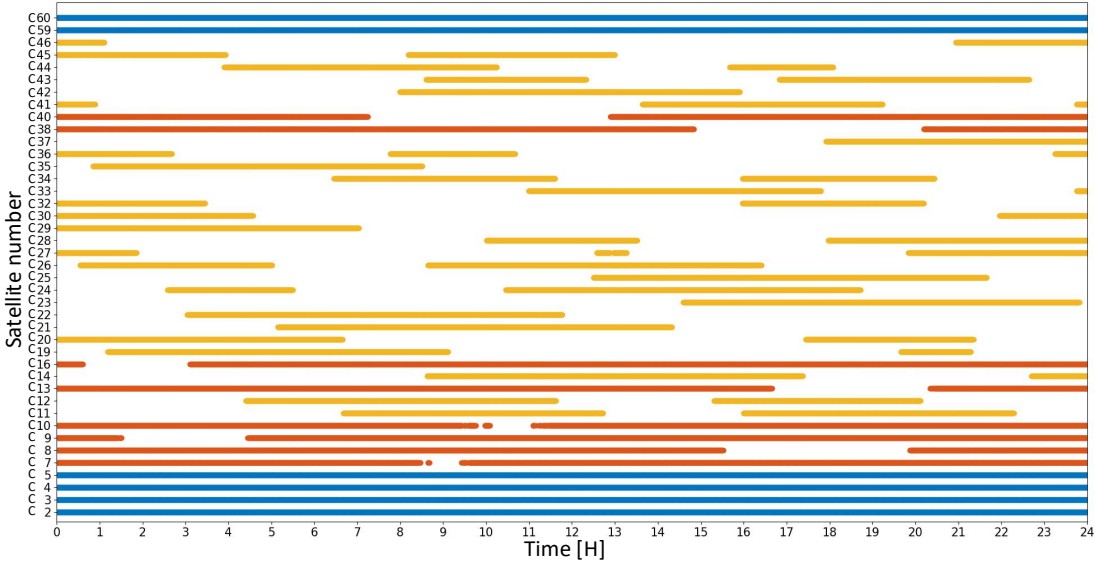

**Figure 5.** China regional SCB data coverage period.

As can be seen from Figure 5 and Table 2, most of the SCB products of BDS satellites have been computed in China. The relative position of the GEO satellites to the ground station hardly changes, so the daily SCB is completely calculated. There are some defects in the IGSO satellite clock deviation; according to the statistical data in Table 2, the defects are about 5–25%. All MEO satellites can be observed in China and about 17–51% of clock bias in the epoch can be calculated every day. Most of satellites signals can be received about 40% epochs of a day. BDS-2 satellites have less vacancy than BDS-3 satellites. This is because the BDS-2 satellite positioning system was originally designed to serve the Asia-Pacific region and the trajectory of its sub-satellite points covered mostly in China, with relatively complete station observations.

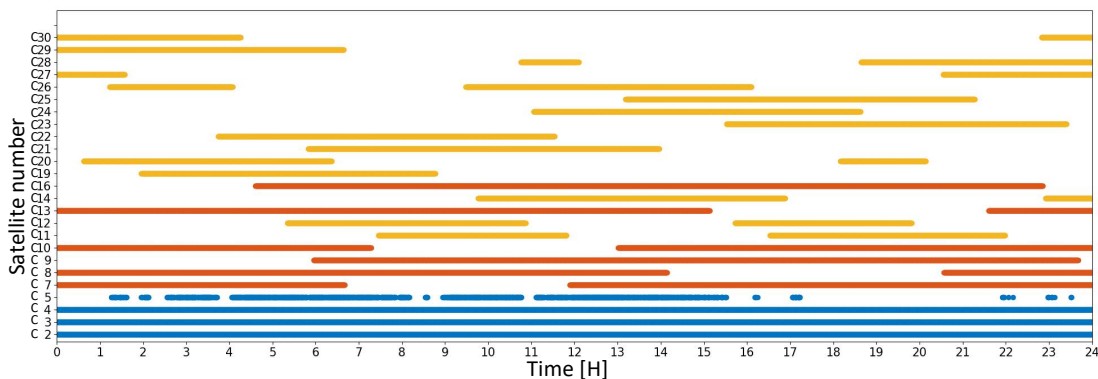

**Figure 6.** North China regional SCB data coverage period.

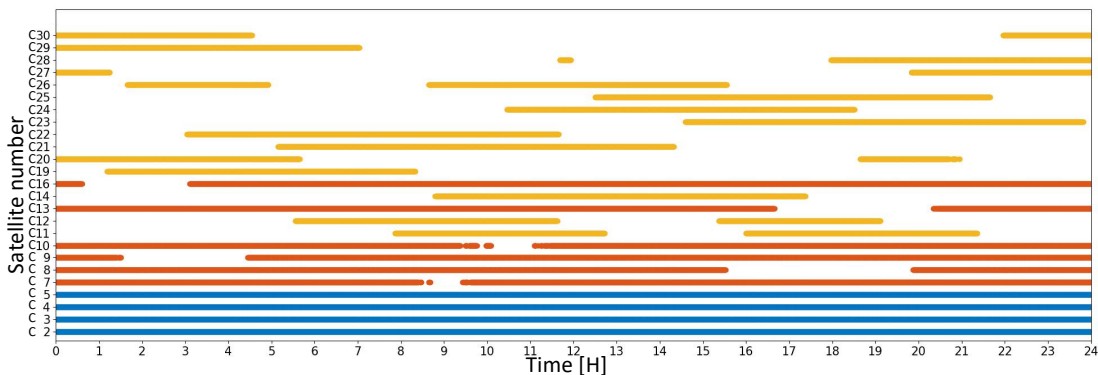

**Figure 7.** South China regional SCB data coverage period.

Figure 4 shows that the stations in the North China region are distributed in the mid-latitude area north of 34 °N. However, the GEO satellites are above the equator, causing the poor signal intensity of some GEO satellites with low altitudes, resulting in the discontinuous observation of C05 in Figure 6. Due to the western position of the projection point of the satellite on the ground of C59 and C60, the observation stations are concentrated in the east of China, which also leads to the lack of observation of the C59 and C60 satellites. Some IGSO satellites (C38–C40) and MEO satellites (C32–C37, C41–C46) of BDS-3 provide services to the Western Hemisphere. These satellites cannot be observed by stations in North China, so the SCB data is missing. It can be seen from Table 3 that about 21–41% of the observed satellites can be used for SCB calculation every day. Table 4 shows that the SCB calculated every day in this region is smaller than that in China. The missing epoch of MEO satellites accounts for about 2–15% of the total epoch per day, and the reason for the lack is mainly due to the smaller distribution area of survey stations. The missing epoch of IGSO satellites accounts for about 9–18% of the total epoch per day; the main reason for the lacking is the higher latitude.

It can be seen from Figure 4 that the stations in South China are distributed in low-latitude areas south of 28 °N and the GEO satellite is above the equator, with good observation conditions. The longitude range of the station is similar to that of the North China region in Figure 3, and the solution of C59 and C60 is also lacking. The lack of calculation data for IGSO and MEO satellites is similar to that of the North China region, indicating that whether the calculation of clock products of IGSO and MEO satellites in China is null is closely related to longitude. For the IGSO and MEO satellites whose solution results are not null, it can be seen from Tables 5 and 6 that the daily coverage rate of clock products in South China is higher than that in North China. The daily coverage rate and period of clock products of some stations in South China are the same as the clock product of the China region, which can also be seen in Figure 7.

**Table 5.** Daily coverage of regional SCB data in South China.

| GEO | | IGSO | | | | MEO | |
|---|---|---|---|---|---|---|---|
| C01 | 0.00% | C06 | 0.00% | C11 | 42.71% | C24 | 33.61% |
| C02 | 100.00% | C07 | 95.73% | C12 | 41.01% | C25 | 38.19% |
| C03 | 100.00% | C08 | 81.94% | C14 | 35.80% | C26 | 42.50% |
| C04 | 100.00% | C09 | 87.67% | C19 | 29.83% | C27 | 22.60% |
| C05 | 100.00% | C10 | 93.75% | C20 | 32.64% | C28 | 26.28% |
| C59 | 0.00% | C13 | 84.55% | C21 | 38.26% | C29 | 29.31% |
| C60 | 0.00% | C16 | 89.65% | C22 | 35.94% | C30 | 27.50% |
| | | C38–C40 | 0.00% | C23 | 38.47% | C32–C37 | 0.00% |
| | | | | | | C41–C46 | 0.00% |

**Table 6.** Comparison of daily coverage of SCB data in each region.

| China Clock Minus South China Clock | | | | | | North China Clock Minus South China Clock | | | | | |
|---|---|---|---|---|---|---|---|---|---|---|---|
| MEO | | | | IGSO | | MEO | | | | IGSO | |
| C11 | 8.82% | C25 | 0.00% | C06 | 0.00% | C11 | −1.70% | C25 | −4.41% | C06 | 0.00% |
| C12 | 9.20% | C26 | 8.58% | C07 | 0.07% | C12 | −0.83% | C26 | −3.02% | C07 | −17.43% |
| C14 | 6.15% | C27 | 5.21% | C08 | 0.00% | C14 | −1.56% | C27 | −1.60% | C08 | −8.68% |
| C19 | 10.10% | C28 | 13.54% | C09 | 0.00% | C19 | −1.39% | C28 | 1.63% | C09 | −13.79% |
| C20 | 11.42% | C29 | 0.00% | C10 | 0.28% | C20 | −0.35% | C29 | −1.60% | C10 | −17.60% |
| C21 | 0.00% | C30 | 0.10% | C13 | 0.00% | C21 | −4.31% | C30 | −4.83% | C13 | −11.46% |
| C22 | 0.49% | C32–C37 | 32.05% | C16 | 0.00% | C22 | −3.40% | | | C16 | −13.58% |
| C23 | 0.00% | C41–C46 | 29.48% | | | C23 | −5.59% | | | | |
| C24 | 13.02% | | | | | C24 | −2.01% | | | | |

In this section, we make statistics on the number of satellites that can be calculated for SCB products in each region (hereafter referred to as the number of observable satellites); the results are shown in Figure 8. As can be seen from Figure 8, in the North China region (middle-latitude region), the number of observable satellites is less than the South China region (low-latitude region). This confirms the conclusion in Table 6 that the stations in the low-latitude region contribute more to the calculation of SCB, and more visible satellites can be observed. More than 10 satellites in the three regions can be calculated to reach the requirements of precise point positioning.

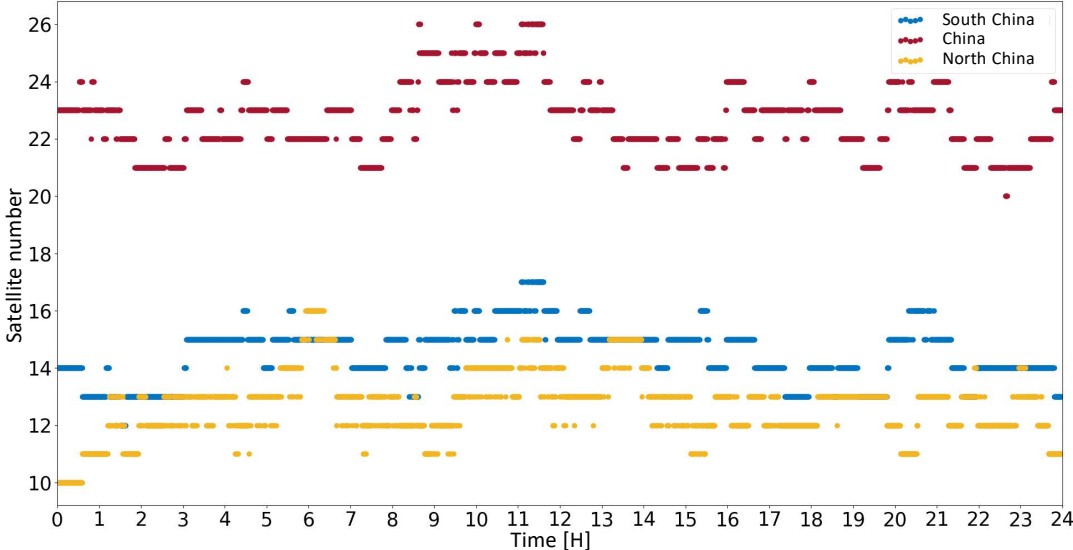

**Figure 8.** Regional SCB data's solvable satellites.

### 3.2. Accuracy of Regional Clock Products

In this paper, the global and regional SCB are calculated by the fixed orbit. The study time is 214–243 days in 2021. The global SCB product is used as the reference, and the standard deviation (STD) of the double-difference SCB of each satellite within the study time is calculated to evaluate the accuracy level of satellite clocks in each region. In the calculation of the double difference, C14 is used as the reference satellite.

It can be seen from Figure 9 that the accuracy level of SCB products is affected by the satellite orbit; the order from high to low is MEO, IGSO, and GEO. The accuracy level of SCB products is also affected by the station region. The accuracy of China regional SCB products in each region is less than 0.7 ns for the GEO satellite, 0.3 ns for the IGSO satellite, and 0.2 ns for the MEO satellite. The accuracy of SCB products in the North China region is lower than that in China. The STD of the GEO satellite is within 0.7 ns, the IGSO satellite is within 0.3 ns, and the MEO satellite is within 0.2 ns. In the regional SCB, the accuracy of clock products in South China is lower than that in the North China region. The STD of GEO satellite clock products calculated in the South China region is within 0.7 ns and that of IGSO and MEO satellite clock products is within 0.3 ns. The stations in South China have lower latitudes and are greatly affected by the ionosphere. Although the ionospheric elimination algorithm is used in this experiment, the active ionospheric variation will still affect the float ambiguity resolution, cycle slip detection, and the quality of ionospheric elimination, resulting in the instability of SCB observation data in the South China region, and the accuracy is worse than that in the North China region.

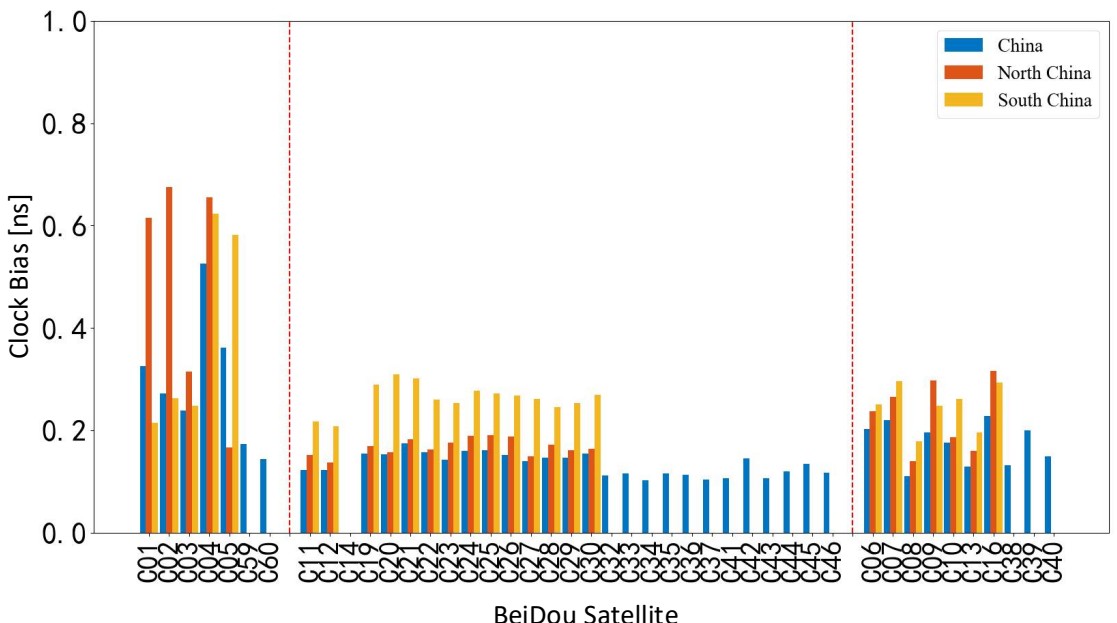

**Figure 9.** The accuracy of SCB in different regions.

### 3.3. Regional Influence Deviation of Clock Product

The phase, frequency, and frequency drift of the satellite clocks are obtained by quadratic term fitting to the global clock products calculated in this paper. The theoretical value of the satellite clock is calculated by Equation (12), and the difference is made with the calculated regional satellite clock product, obtaining the mix value of noise and regional influence. Figures 10–12, respectively, show the influence of the China region, North China region, and South China region on SCB calculation (cyan for GEO, red for IGSO, green for MEO). It can be seen from the figure that the level of the mixed value of noise and regional influence is $10^{-7}$ s. In Section 3.2, we have made statistics on the accuracy of regional clock products and know that the accuracy of regional clock products is within $10^{-9}$ s. Based on the above result, it can be seen that noise accounts for a relatively small proportion

of the mixed value. We believe that the influence of noise on the region can be ignored for analysis. The regional influence is reflected in the phase offset of the satellite clock calculation. It absorbs some atmospheric parameters, and the influence on the satellite clock bias of all satellites has obviously similar regional characteristics, so it is named as the Region Effect Bias (REB) below. Different from the global SCB product, the regional SCB product with REB can reflect the sudden change of atmospheric delay parameters over the studied area and has a good enhancement effect on the regional PPP.

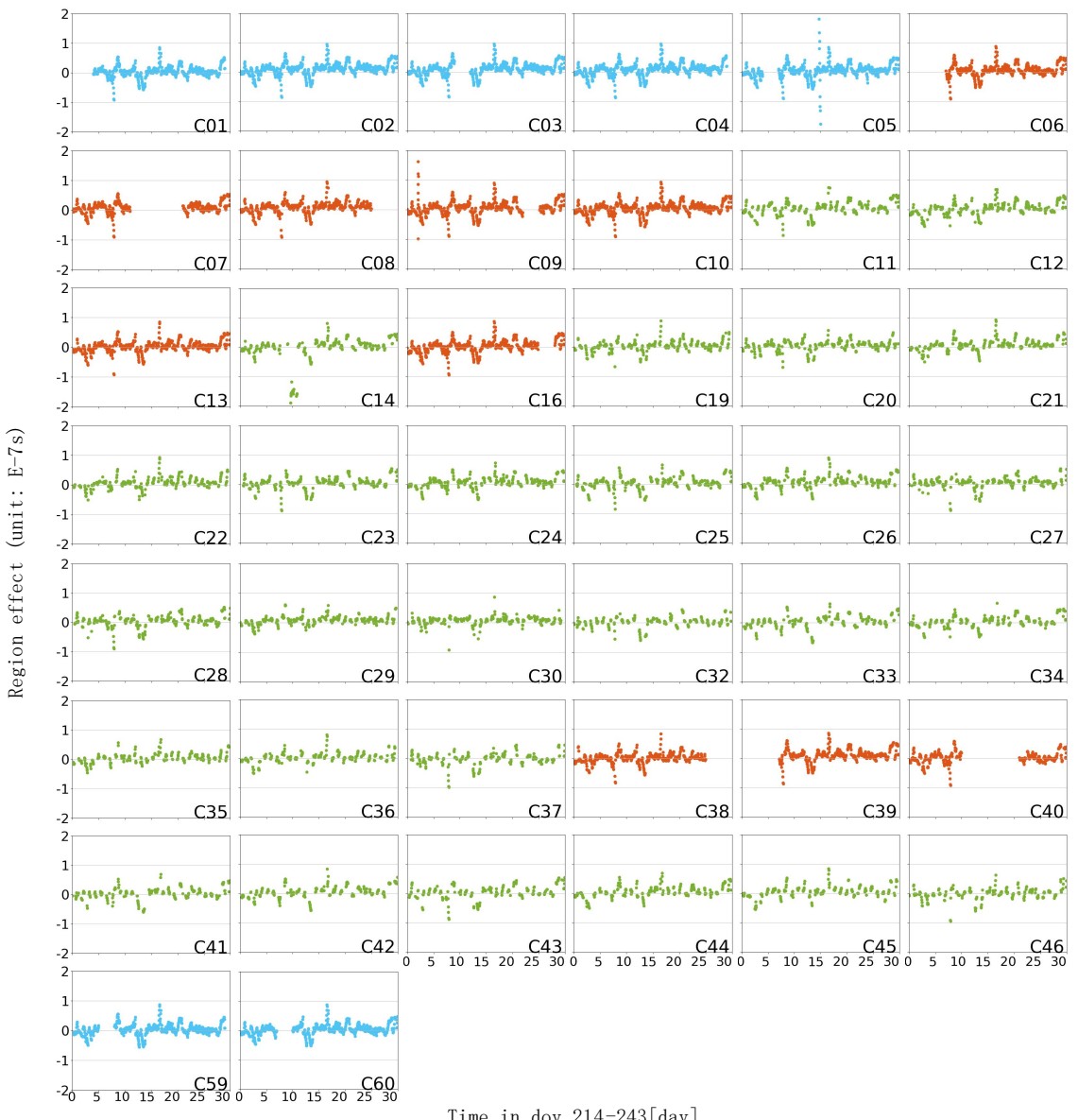

**Figure 10.** China region REB ($10^{-7}$ s).

It can be seen from Figures 10–12 that the time series curves of regional influence bias in the same region are similar. Figure 10 clearly shows that the REB of all solvable SCB products has the same time series characteristics. Although there are many missing epochs in Figures 11 and 12, this situation can also be observed. This indicates that the REB in SCB calculation is related to the distribution region of the station, the REB of the same group of regional networks has the same fluctuation trend on all satellites. Observed in the time series, all satellites in the same area have similar trends in the REB variation, which is irrelevant to the orbits of the satellites. This shows that the REB value of the satellite

clock has a strong relationship with the region. The influence factors, such as atmospheric parameters over the region, lead to the rapid change of the REB value in the entire region, and the overall variation range is at the level of $10^{-7}$ s. In the same area, there are slight differences between satellites. It can be seen that the MEO satellites, which are shown as green, are obviously sparser than the IGSO satellites in red, indicating that there are more missing epochs in the clock error calculation of the MEO satellites. Additionally, BDS-2 satellites are denser than BDS-3 satellites, indicating that BDS-3 satellites are missing more epochs. The above two conclusions confirm the discussion of the completeness of satellite clocks in different orbits in Section 3.1. Moreover, in the same area, the fluctuation range of MEO satellites is smaller than that of IGSO and GEO orbiting satellites, and its number of the outliers are less, which is also due to the better orbit determination effect of MEO satellites.

During the study period, the regional influence bias and the range of change in the whole area of China are small. The REB value in the North China region is larger than that in the whole of China, and some values fluctuate significantly. The REB values of China and the North China region are mostly in the range of $\pm 0.5 \times 10^{-7}$ s, whereas the REB values of the South China region fluctuate sharply in the range of $\pm 1 \times 10^{-7}$ s. This shows that the area with a larger range or with a higher latitude has less influence, and the result of this REB value is the SCB parameter absorbing the regional systematic error. However, the ionospheric activity in the lower latitude regions is more active, and it has a greater impact on the regional SCB. The above results provide a reference for station selection in the application of regional clock products; that is to say, when the distribution range of the stations is small, in order to reduce the regional influence, it is appropriate to consider selecting the higher latitude stations to calculate the SCB.

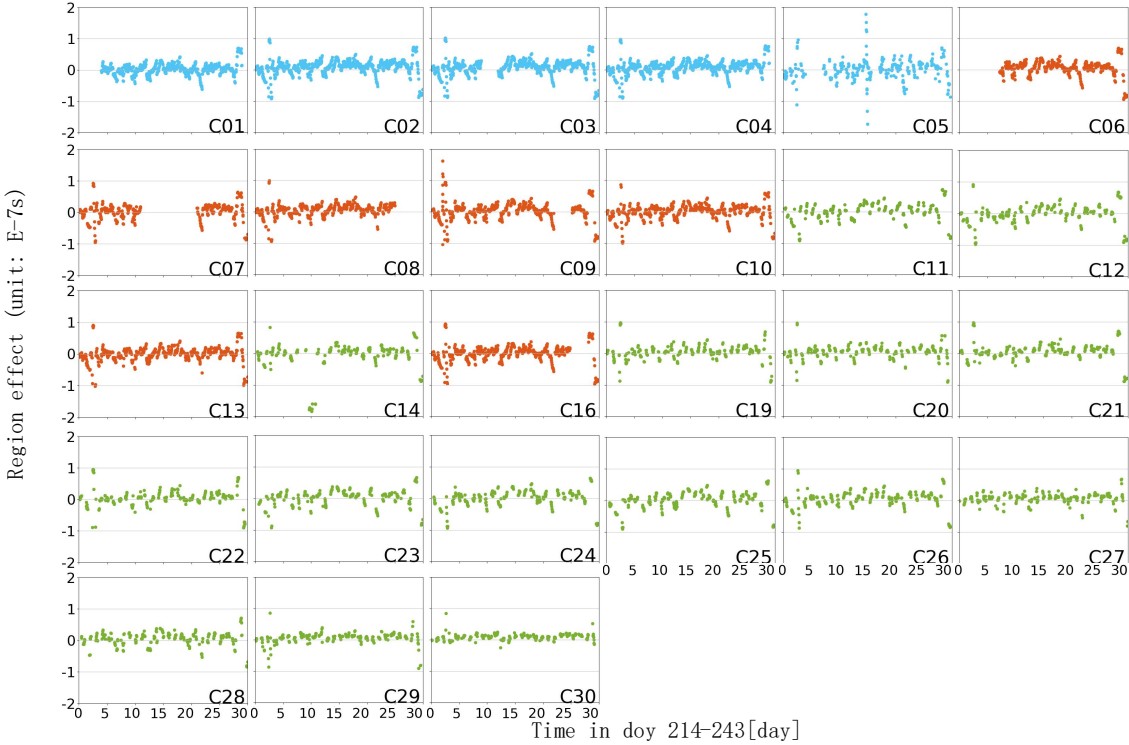

**Figure 11.** North China region REB ($10^{-7}$ s).

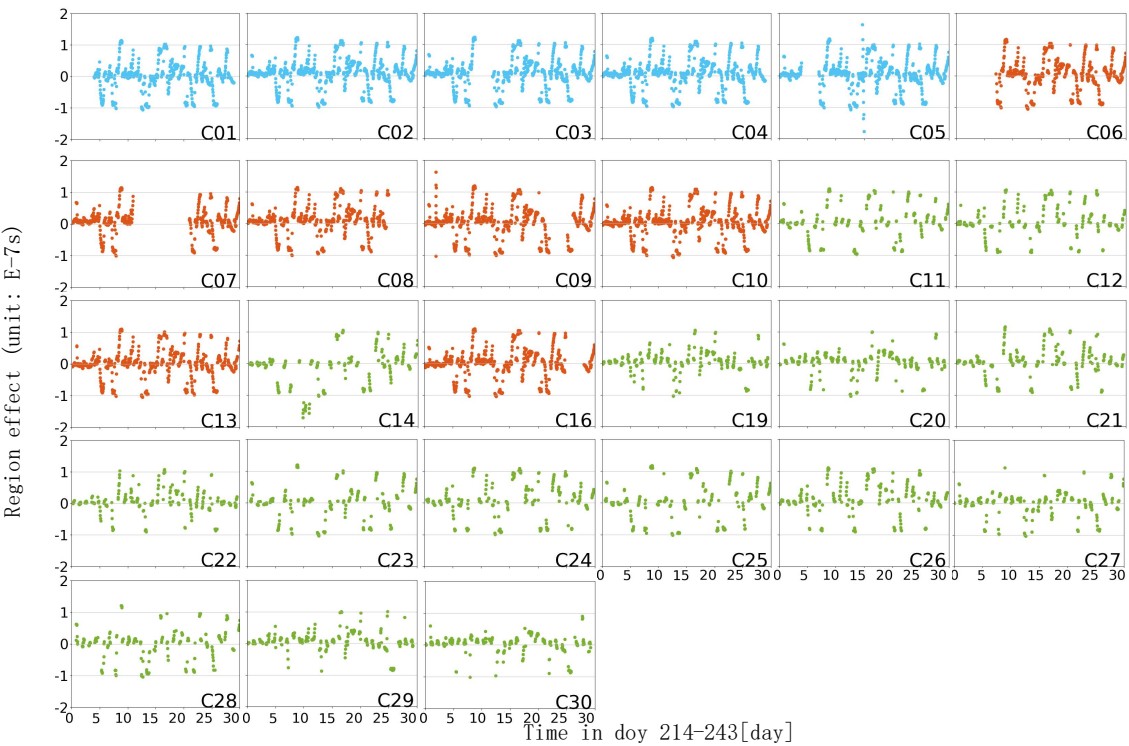

**Figure 12.** South China region REB ($10^{-7}$ s).

## 4. PPP Analysis of Regional Clock Product

We selected IGS stations in the Asia-Pacific region and national stations which had not participated in the calculation in each region to verify the PPP performance of the regional satellite clock product, and the selected stations can receive BDS-2 and BDS-3 signals and use the B1B3 signal for PPP processing. The distribution of IGS stations and national stations is as Figure 13.

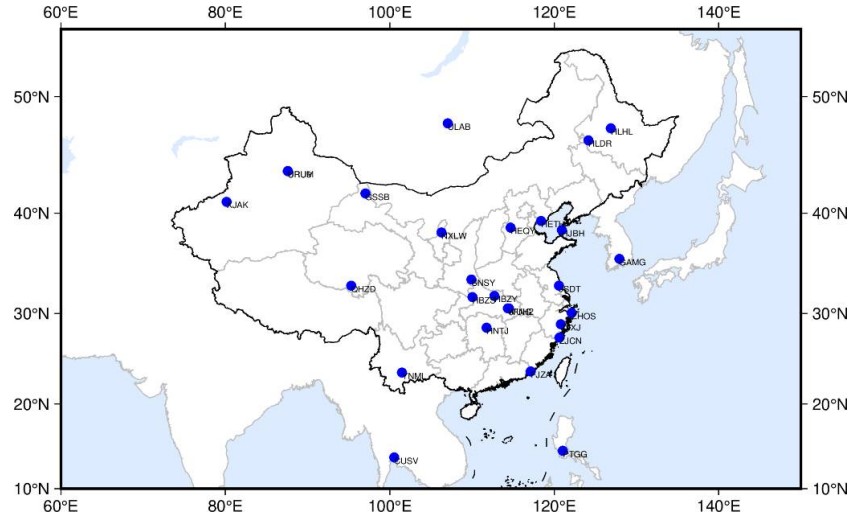

**Figure 13.** PPP station distribution.

The satellite clocks calculated by the global, Chinese, Northern, and Southern China regional stations are used to conduct PPP experiments for the selected stations on days 214–220 of 2021. The first epoch of 20 consecutive epochs when the error is less than 10 cm is used as the convergence time to count the post-convergence accuracy. The seven-day averages of Root Mean Square Error (RMS) after convergence at different IGS stations are counted in Figure 14; the RMS in E, N, and U directions are combined, i.e.,

$$\delta = \sqrt{\delta_E{}^2 + \delta_N{}^2 + \delta_U{}^2} \tag{13}$$

In Equation (13), $\delta$ represents the three-dimensional (3D) RMS in the space; $\delta_E$, $\delta_N$, and $\delta_U$ represent the RMS error in the direction of E, N, and U. It should be noted that in the RMS calculated by the Equation (13), due to the large value of RMS in the U direction, small values in the E and N directions will be submerged, making the overall RMS statistical data seems large.

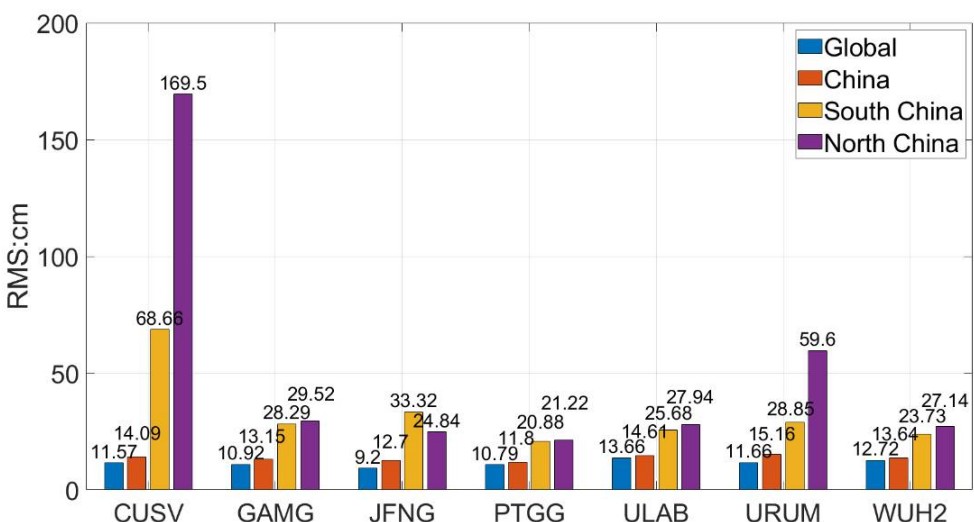

**Figure 14.** Effects of Different Regional Clocks on Different IGS Stations.

From Figure 14, it can be seen that the PPP results of the stations near China solved by the global SCB product have the same accuracy, and the 3D accuracy is at the level of 10 cm. The RMS error of the PPP calculated by the small size regional clock product is higher than that of the global product, which indicates that the reduction of the calculation range of the regional clock has a negative impact on the PPP. The difference between the accuracy of the Chinese regional clock solution and the global solution is insignificant, and they are both at the same level. This indicates that the accuracy of PPP experiments is very close to that of the global clock products if only the SCB products calculated by the national stations in the Chinese region are used. When comparing within the Chinese region, the PPP accuracy of CUSV using the regional clocks in South China in the figure is much higher than that in North China. This phenomenon combined with Figure 8 shows that when performing PPP, the accuracy level of the regional stations calculated far from the clock product increases rapidly with the number of observable satellites. However, the stations closer to the center of the study area, such as PTGG, ULAB, and WUH2, have a larger base of observation satellites, and the continued increase in the number of observation satellites will reduce the contribution to the improvement of PPP accuracy. However, the low-latitude regional SCB product has poor accuracy, as shown in Figure 9, and a large deviation from its regional influence, which prevents further improvement of the accuracy. It will even reduce the PPP accuracy of some stations such as JFNG.

The combination of Figures 14 and 15 shows that the accuracy has reached the level of global clock accuracy when the corresponding satellite clocks are used for PPP in the regional range calculated by the same regional SCB. The stations QHZD, XJAK, YNML, and NXLW in the western region have larger errors, whereas the errors in the southern region are smaller. This phenomenon is similar to the CUSV in Figure 14. This is because the number of satellite observations in the northern region is smaller and the number of satellite observations of the stations far from the satellite clock solution area decrease, and the PPP result accuracy becomes worse. The 3D accuracy of other stations in the North and South China clock calculation regions is at the same level as the global clock, indicating that the SCB products in the smaller mid-latitude range are not suitable for extending the positioning area excessively due to the low number of satellite clock calculations.

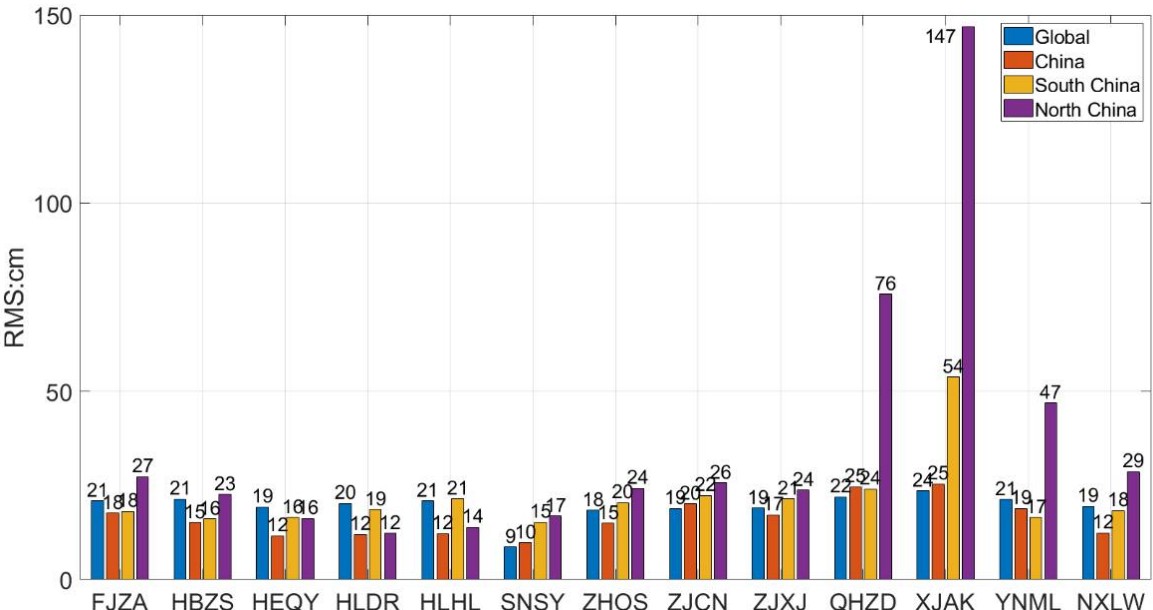

**Figure 15.** Effects of Different Regional Clocks on Different National Stations.

Figures 16 and 17 show the time series of the jump in the PPP results. From the regional clocks of China and North China in Figure 16, in kinematic mode, it can be seen that the PPP results by the regional SCB of China and the global SCB are at the same level, i.e., centimeter level. As shown in kinematic mode, the PPP accuracy of the regional clocks in South China and North China is at the centimeter level in most of the epochs, and there are jumps in the kinematic PPP result sequences of CUSV and URM stations, which shows that some periods after convergence diverge again, and the regional clocks in mid-latitude regions have more jumps and large fluctuations in PPP of stations in low-latitude regions. The regional clock in South China has the same effect on the PPP results of FJZA and CUSV stations, both of which are in the southeast corner of the Chinese region, indicating that the effect here comes from the REB, whereas the jump in the same region using the regional clock in North China has the same obvious consistency as that of the regional clock in South China, and the REB of the regional clock in North China has no obvious effect on the PPP. In the North China region, the low number of observations leads to a decrease in the number of satellite observations in the region of some stations far from the calculation region affecting the accuracy, which leads to the jump in the positioning results of the North China regional clock and affects the positioning accuracy of the regional clock products. In the kinematic mode, the PPP results of the previous epoch are not transferred to the next epoch for iteration to improve the convergence results, whereas in the static PPP mode, the process noise in the state transfer matrix is 0. From the results of static mode positioning, it can be seen that the positioning accuracy of the regional clock reaches

the level of the global clock, but the convergence time is significantly longer. From the comparison between Figures 16 and 17, it can be seen that the static mode can effectively eliminate the jump caused by the steep drop in the number of observation satellites in one day, but from the positioning results of the South China clock for FJZA on CUSV, the effect of REB cannot be eliminated completely. Moreover, the convergence time generally reaches more than 2h, which is obviously inferior to the convergence time of the global clock.

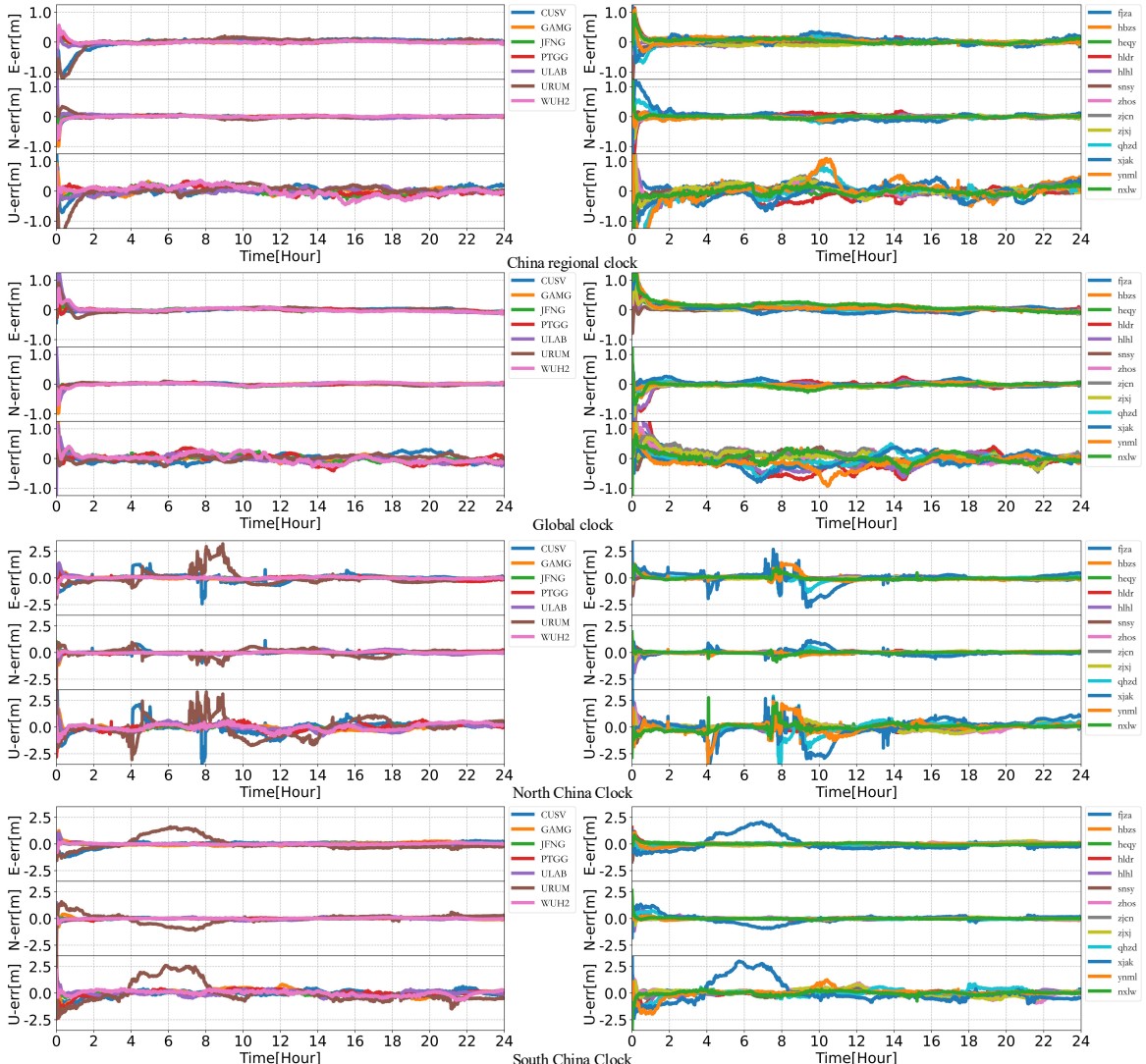

**Figure 16.** Kinematic PPP Time Series Plots of Different Regional Stations on Day 216 of 2021.

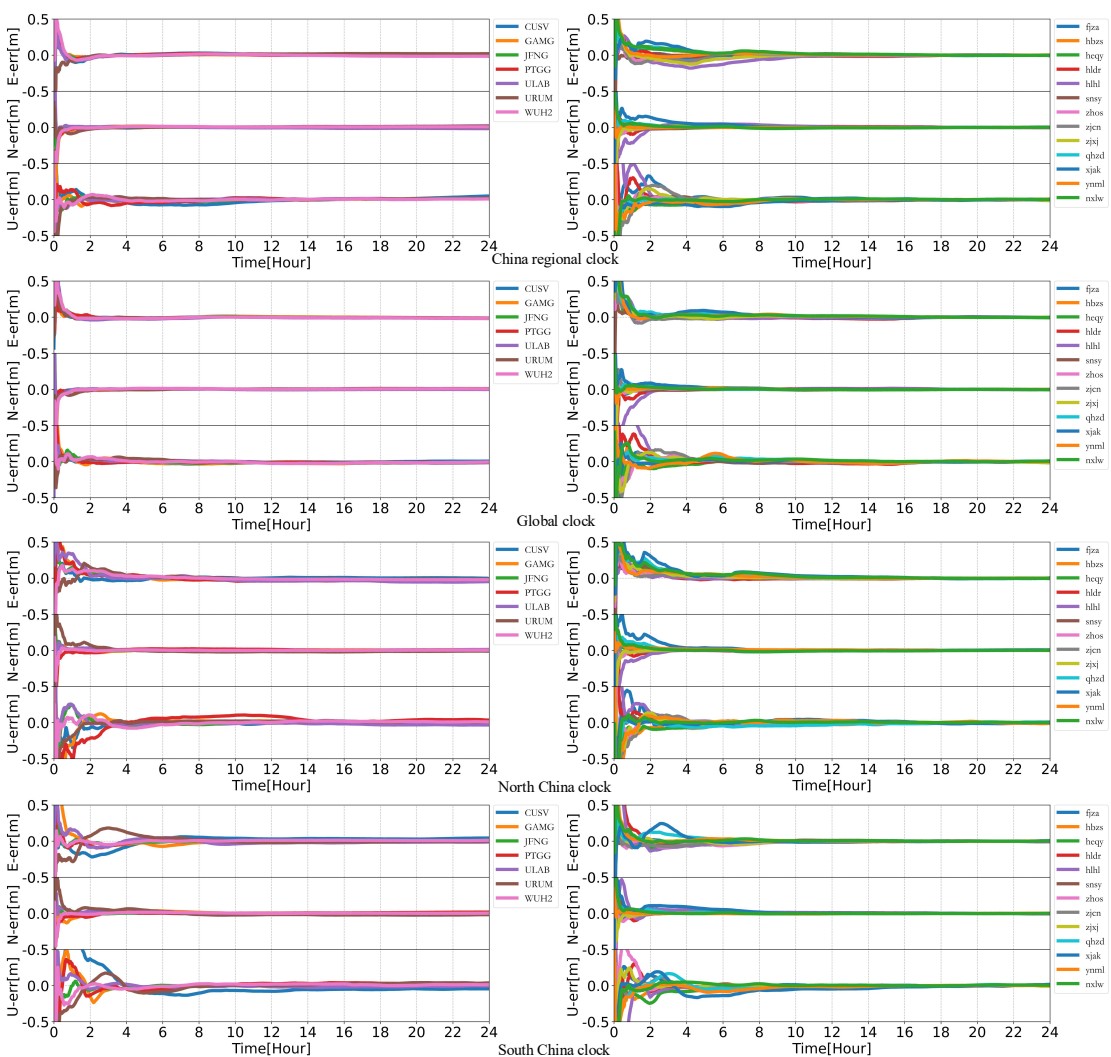

**Figure 17.** Static PPP Time Series Plots of Different Regional Stations on Day 216 of 2021.

## 5. Discussion

We have mentioned the problems of accuracy and regional influence caused by station area and latitude in Sections 3 and 4, and we introduce the concept of regional effect bias (REB). It is found in Section 4 that the stations distributed in the same area have the same fluctuations in the kinematic mode. We believe that the influence of these regional station selections on the BeiDou SCB products comes from the spatial correlation errors such as atmospheric parameters absorbed over the small regional stations. If the research scope is larger, as shown in Figure 2, the difference in spatial correlation error between global and region is small, so the deviation of regional influence is reduced. In Section 3.1, we know that the regional clock has many breakpoints. The calculation after each breakpoint needs to be initialized with the pseudo-range. Therefore, in theory, the regional clock must have a lot of errors due to initialization, and the positioning results should not be as good as the global SCB. However, the positioning accuracy of the regional clock is higher than that of the global SCB; this phenomenon is due to the effect of the SCB parameters absorbing regional system deviation. When the regional SCB calculating, the corresponding range fluctuates when the atmospheric environment changes abruptly and will be absorbed by the regional SCB. These mutations are the main factor leading to the loss of precision in PPP. The atmospheric parameters absorbed by the regional SCB products can offset the effects of these sudden changes during PPP, thereby improving positioning accuracy. Otherwise, some papers also provide another hypothesis that when using globally distributed stations, satellites

are always directly above the visible station [33]. Therefore, the clock estimated from the global network cannot compensate for the orbital error of the tangential component, which may also lead to the degradation of the positioning performance of the global network.

Another important question is whether REB will affect the precision statistics of SCB; because Equation (9) of the regional clock should include the regional effect bias REB, it may affect the statistics and analysis of the SCB accuracy after the double-difference. However, according to the conclusions obtained from Figures 10–12 in Section 3.3, the fluctuations of the REBs of each satellite clock calculated in the same area are the same, and the trend term of REB will be greatly weakened after the difference between satellites. The STD level of the final obtained SCB statistics in Figure 9 is consistent with the normal situation, so the SCB products calculated by the inter-satellite difference method adopted in this paper are effective.

## 6. Conclusions

In this paper, the MD method is used to solve the SCB of three regions, China, North China, and South China. The characteristics of the regional clocks are analyzed by using the global clock as the reference. The regional effect bias REB is introduced to measure the deviation of the calculated regional clock from the true value, the accuracy of the regional satellite clock is evaluated, and the PPP is used to verify the regional clock positioning results. The conclusions are as follows:

1.  Due to the different designs of the systems, the data completeness of the BDS-2 satellite over the China region is better than that of the BDS-3 satellite. The observation number of the regional BeiDou clocks is ranked from highest to lowest in the Chinese region, the South China region, and the North China region. The decrease of station area has a great influence on the observation of the MEO satellite, whereas the increase of latitude has a great influence on the observation of the GEO and IGSO satellite, which leads to the loss of these regional SCB products, respectively. The contribution of low-latitude stations to regional SCB calculation is higher than that of the mid-latitude stations, which indicates that to increase the number and duration of observation satellites, it is necessary to increase the number of observation stations in the low-latitude region. The available satellite number of SCB in each region satisfies the requirements of the PPP experiment.

2.  From the STD of regional clock products, it can be seen that the accuracy of the BeiDou satellite regional clock is from high to low in the China region, North China region, and South China region from high to low. The STD of GEO satellites in all regions are less than 0.7 ns, and that of IGSO and MEO satellites are less than 0.3 ns. The accuracy of SCB products in South China is worse than that in North China, which is due to the low latitude of the stations and the influence of more serious ionospheric errors. Although the ionospheric-free algorithm is used in this experiment, the ionospheric scintillation caused by the active ionosphere will still affect the ambiguity float solutions and the quality of ionospheric elimination, resulting in the instability of SCB observation data in the South China region, and the accuracy is worse than that in the North China region. Therefore, the influence of the ionospheric layer on the accuracy of SCB can be weakened by appropriately choosing the station with higher latitude.

3.  In this paper, the regional effect bias REB is introduced to analyze the influence of different regions on the clock error product. For the BeiDou system, the regional influence bias of the same group of regional networks will lead to a similar offset sequence for all satellite clock errors calculated by this group of regional networks, and the REB value is at the level of $10^{-7}$. Among them, the fluctuation range of MEO's offset series is smaller, and there are fewer outliers. During the study period, the larger regional REB deviations and changes are smaller. The regional influence of South China with lower latitude is stronger when the area is similar and leads to worse PPP accuracy. This shows that the regional influence between the larger area and the higher latitude area station-satellite is small. When the distribution range of

the station is small, for weakened regional influence, the higher latitude station can be appropriately considered to calculate the SCB.

4.  When using the different regional SCB products calculated in this paper for PPP, the amount of data for SCB-computing using stations in the mid-latitude region is maintained at a low level, resulting in a more serious decrease in the accuracy level of the stations at slightly distant locations near the clock product solving area. When using mid-latitude for regional positioning, the working area for positioning should be controlled as much as possible.

5.  The PPP results of the Chinese regional clock and the global clock are at the same level, but the convergence speed of the regional clock is generally inferior to that of the global clock. In the regional mode, the low-latitude clocks are affected by the region, and the mid-latitude clocks are affected by the regional influence, whereas the mid-latitude clocks have a jump in accuracy due to the low amount of observed data. The static mode can improve the fluctuation due to the small amount of data, but it does not completely eliminate the effect of REB.

**Author Contributions:** Conceived the idea, H.W. and W.L.; designed the software, collected the data, and analyzed the experimental data, W.L. and P.L.; collected the related resources and supervised the experiment, W.L. and H.W.; proposed the comment for the paper and experiment, H.M., Y.R., B.L., Y.C. and H.W.; investigation, W.L. All authors have read and agreed to the published version of the manuscript.

**Funding:** This research is supported by the Key Project of China National Programs for Research and Development (No. 2022YFB3903902; No. 2022YFB3903900), the Wenhai Program of the S&T Fund of Shandong Province for Pilot National Laboratory for Marine Science and Technology (Qingdao) (No. 2021WHZZB1000, No. 2021WHZZB1005), the National Natural Science Foundation of China (No. 42274044; No. 41874042), the State Key Laboratory of Geo-Information Engineering and Key Laboratory of Surveying and Mapping Science and Geospatial Information Technology of MNR, CASM (No. 2022-01-09; No. 2021-01-01), and the Scientific Research Project of Chinese Academy of Surveying and Mapping (No. AR2101; No. AR2203; No. AR2214).

**Institutional Review Board Statement:** Not applicable.

**Informed Consent Statement:** Not applicable.

**Data Availability Statement:** Not applicable.

**Acknowledgments:** The authors acknowledge IGS (International GNSS Service) for GNSS observations and CODE (Center for Orbit Determination in Europe), WHU (Wuhan University), and GFZ (Deutsches GeoForschungsZentrum) for GNSS products. Additionally, some of the figures are produced by GMT (Generic Mapping Tools).

**Conflicts of Interest:** The authors declare no conflict of interest.

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
