# Peer review of "Analysis of Regional Satellite Clock Bias Characteristics Based on BeiDou System"

_remotesensing, doi:10.3390/rs14236047_

Round 1

Reviewer 1 Report

Overall the paper presents some problems:

1) The state of the art should be investigated more deeply

2) It requires some editing due to lack of clarity in its content.

3) In order to make the results more relevant, I would suggest to process more data.

I have also the following comments:

* Line 61 - There is a typo. Phrase instead of phase.

* Line 72 - RT-PPP was already defined in line 50

* Lines 77-81 - Rephrase for better clarity

* Lines 98-109 - Rephrase for better clarity

* Why the geometric term does not appear in equations 3 and 4? Also the delta_t are not defined in the following explanation.

* Lines 127-128 - Here you refer to equation 4, which is the time differenced phase measurement. However in eq. 5 you rewrite eq. 3. Which equation are we talking about here?

Lines 130-131 - Explain why the last two terms in eq. 5 can be replaced by the clock bias that has been calculated in the previous day.

Line 137 - Is it really eq. 4?

Eq 7 - What's the need to rewrite this? Is it not essentially the same as eq. 5?

Lines 144-145 - What's the daily of global satellites?

Line 145 - What do we mean by integrity in this paper?

Line 153 - Explain better what the initial clock deviation is.

Line 154 - I am assuming that R_sa is the error that correlated with the radial orbit component. This should be better explained.

Line 171-172 - Why is R_DaDs 0?

Eq. 11 - L_i is normally used in the literature to identify the phase measurements, but here it is used for the clock bias data. I think it may generate some misunderstanding. Please consider rewriting this equation using different symbols.

Lines 178-183 - This is not very clear. Please rephrase for better clarity.

Table 1 - One of the parameters mention the ocean tides, but it actually includes also solid earth and pole tides. In the parameter column I would write simply "Tides".

Figure 1 - How's consistency defined here? Please explain.

Section 3.1 - Why there is a concern for the limited visibility of satellites (and therefore clocks) in a regional network? How would this be a problem for users in that region? Please explain.

Lines 209-210 - Rephrase for better clarity.

Lines 210-211 - What do you mean by integrity in this section? What are the integrity requirements for PPP? Can you provide some numbers?

Table 2, 3, 5 and 6 - I don't understand these tables. I was assuming that the SVIDs where sorted by satellite type (GEO, IGSO, MEO), but I don't get why we have the same SVIDs in multiple columns.

Figure 5, 6 and 7 - Please specify what the different colors mean.

Lines 222-223 - What prevents the clocks to be computed?

Lines 224-225 - Rephrase for better English.

Lines 227-228 - Rephrase for better English

I would use more data for the analysis. Is it possible to process more days?

Lines 301-313 - Explain better what REB is and why it is an interesting and useful indicator.

Lines 321-323 - Argument this better.

Author Response

We appreciate all experts for their criticism and corrections on this article.

For the convenience of experts to review:

we denote the expert opinions in black italics and numbered sequentially. 

Our explanations are denoted in blue italics below the corresponding expert opinions. 

Our replacement of the original text is shown in blue non-italics and marked with the position before and after the modification.

Reviewer 2 Report

The paper presents an interesting work. I do not have specific concern related to the work. I can only suggest to verify the usage of the acronyms some of them are repeated. In addition I suggest to not use system after GNSS because the acronym already contains the word system. Most of the figures are clear and easy to read. I suggest to modify figures 10, 11 and 12 presenting one or two satellites and reporting in the text that similar results are obtained for the other satellites. In this way the figures will be easier to read.

For the literature review, I suggest to include one or two paper on HAS provided by Galileo to complete the techniques presented.

Author Response

Thanks to the expert's review and correction, we checked the grammar and vocabulary, removed the system after GNSS, and removed repeated definitions of acronyms, used the acronym for more frequently used words like satellite clock bias (SCB). Please reference the attachment for more details.

For the convenience of experts to review:

we denote the expert opinions in black italics and numbered sequentially. 

Our explanations are denoted in blue italics below the corresponding expert opinions. 

Our replacement of the original text is shown in blue non-italics and marked with the position before and after the modification.

Round 2

Reviewer 1 Report

A few other comments

Lines 83-89: Galileo HAS Level 2 service offers higher level of accuracy wrt the HAS Level 1 thanks to the regional atmospheric data covering Europe. However, it is stated in the ICD that the orbit clock and bias corrections are the same as the Level 1, hence global corrections. I would suggest the authors to reference papers  looking into CLAS correction service from QZSS satellites. This service actually has a full set of local corrections that are different from the global ones.

Comment (19) in the cover letter - I understand the concern here. If a satellite is not visible by any ground stations within the regional network, it is not possible to compute the clock corrections for that same satellite. On the users side however, if they are also within the same region, it is likely that these satellites are not visible as well. So the poor visibility is mainly due to the fact that the satellites are not observable in that region and not by the lack of corrections (i.e. even if the corrections were provided, still the satellite could not be used for positioning). As it is presented in the paper, unless the regional corrections could be used by users outside of that same region, I still don't see the motivation for this analysis.

Author Response

We responded to comments from reviewers and editors. Please see the attachment.
